Corrected: Author correction

# Oxidative rearrangement of (+)-sesamin by CYP92B14 co-generates twin dietary lignans in sesame

Jun Murata [1], Eiichiro Ono[2], Seigo Yoroizuka[3], Hiromi Toyonaga[2], Akira Shiraishi[1], Shoko Mori[1], Masayuki Tera[1], Toshiaki Azuma[1], Atsushi J. Nagano[4,5], Masaru Nakayasu[6], Masaharu Mizutani[6], Tatsuya Wakasugi[3], Masayuki P. Yamamoto[3] & Manabu Horikawa [1]

(+)-Sesamin, (+)-sesamolin, and (+)-sesaminol glucosides are phenylpropanoid-derived specialized metabolites called lignans, and are rich in sesame (*Sesamum indicum*) seed. Despite their renowned anti-oxidative and health-promoting properties, the biosynthesis of (+)-sesamolin and (+)-sesaminol remained largely elusive. Here we show that (+)-sesamolin deficiency in sesame is genetically associated with the deletion of four C-terminal amino acids (Del4C) in a P450 enzyme CYP92B14 that constitutes a novel clade separate from sesamin synthase CYP81Q1. Recombinant CYP92B14 converts (+)-sesamin to (+)-sesamolin and, unexpectedly, (+)-sesaminol through an oxygenation scheme designated as oxidative rearrangement of α-oxy-substituted aryl groups (ORA). Intriguingly, CYP92B14 also generates (+)-sesaminol through direct oxygenation of the aromatic ring. The activity of CYP92B14 is enhanced when co-expressed with CYP81Q1, implying functional coordination of CYP81Q1 with CYP92B14. The discovery of CYP92B14 not only uncovers the last steps in sesame lignan biosynthesis but highlights the remarkable catalytic plasticity of P450s that contributes to metabolic diversity in nature.

[1] Bioorganic Research Institute, Suntory Foundation for Life Sciences (SUNBOR), 8-1-1 Seikadai, Seika, Soraku, Kyoto 619-0284, Japan. [2] Research Institute, Suntory Global Innovation Center Ltd (SIC), 8-1-1 Seikadai, Seika, Soraku, Kyoto 619-0284, Japan. [3] Graduate School of Science and Engineering, University of Toyama, 3190 Gofuku, Toyama 930-8555, Japan. [4] Faculty of Agriculture, Ryukoku University, 1-5 Yokotani, Seta Oe, Otsu, Shiga 520-2914, Japan. [5] JST CREST, 4-1-8 Honcho, Kawaguchi, Saitama 332-0012, Japan. [6] Graduate School of Agricultural Science, Kobe University, Kobe 657-8501, Japan. Jun Murata, Eiichiro Ono, and Seigo Yoroizuka contributed equally to this work. Correspondence and requests for materials should be addressed to M.P.Y. (email: mpyama@sci.u-toyama.ac.jp) or to M.H. (email: horikawa@sunbor.or.jp)

Sesame (*Sesamum indicum*) has been cultivated since antiquity and charred sesame was found in a stratum dated at between 3050 and 3500 B.C. at an Indus Valley Civilization site[1]. Sesame seed oil is a rich source of unsaturated fatty acids, as well as lignans, which are phenylpropanoid-derived plant specialized metabolites chemically defined as monolignol dimers[2,3]. (+)-Sesamin, (+)-sesamolin and (+)-sesaminol glucosides are the major lignans in sesame seed, and have attracted attention for their health-promoting activities. For example, (+)-sesamin is a phytoestrogen, a precursor of the mammalian lignans enterolactone and enterodiol, and exhibits protective activity against breast and prostate cancers[4]. The long-term consumption of (+)-sesaminol is believed to inhibit pathogenic extracellular β-amyloid aggregation associated with Alzheimer disease[5]. Moreover, sesamol, a sesame lignan-related metabolite, has been related to photoprotective effect in UV-B irradiated lymphocytes, and repression of melanin biosynthesis by reducing cAMP-dependent tyrosinase activity[6,7]. Lignans are often observed in the seeds of oil crops, such as olive, safflower, and flaxseed, as well as sesame, and have been implicated to be potent defense phytochemicals against microbes and other plant species[8]. However, in sharp contrast to the wealth of knowledge on the biological activities of lignans in mammals, the physio-ecological roles of lignans in plants remain largely elusive.

The biosynthesis of sesame lignans up to (+)-sesamin is well characterized. The pathway branches out from a common cascade of phenylpropanoids by stereo-selective radical coupling of two molecules of a monolignol, coniferyl alcohol, guided by a dirigent protein[9]. This coupling reaction results in the formation of the (+)-enantiomer of pinoresinol, a central precursor of lignans.

(+)-Pinoresinol is then converted to (+)-sesamin via sequential oxygenation resulting in the formation of two methylenedioxy bridges. This reaction is catalyzed by a single cytochrome P450 monooxygenase CYP81Q1, also known as piperitol/sesamin synthase[10]. In contrast, the terminal metabolism of sesame lignans, particularly (+)-sesamolin and (+)-sesaminol, remain enigmatic, with neither the type of oxygenation nor the putative substrate being well understood (Fig. 1a). This is primarily because (+)-sesamolin biosynthesis requires an atypical oxygen insertion between the furan and aromatic rings. This reaction has been rarely observed other than *Justicia simplex* and *Phryma leptostachya*[11,12]. Through genetic, genomic and biochemical approaches, here we identify a P450 monooxygenase CYP92B14 as the enzyme that is responsible for the oxygenation of (+)-sesamin to form (+)-sesamolin and (+)-sesaminol. Furthermore, the labeling experiments using stable isotopes indicate that the oxygenation involves a novel reaction scheme designated as oxidative rearrangement of α-oxy-substituted aryl groups (ORA). The identification of CYP92B14 completes the biosynthesis of major lignans in sesame, and provides insights into yet to be characterized mechanisms of enzymatic oxygenation in specialized metabolites that possess heteroatoms at the α-position of alkyl-substituted aryl groups.

## Results

**Genetic analysis of sesamolin-deficient accession #4294.** In order to identify enzyme genes responsible for the biosynthesis of (+)-sesamolin, recombinant inbred lines (RILs; F6 generation) of sesame were generated by crossing a (+)-sesamolin-deficient accession #4294 and a (+)-sesamolin-accumulating cultivar

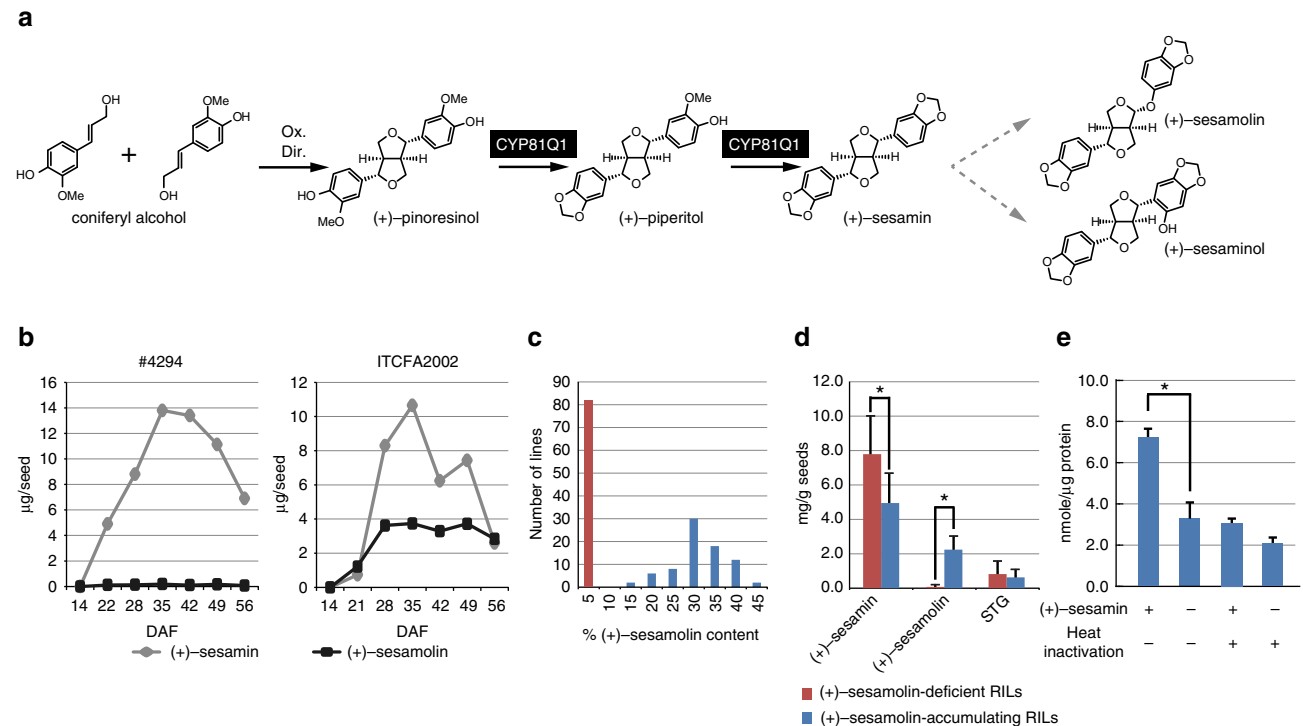

**Fig. 1** Biosynthesis of (+)-sesamolin. **a** Putative biosynthetic pathway of (+)-sesamolin. Ox: oxidant, Dir: dirigent protein, dashed gray arrows; reactions catalyzed in this study. **b** The amounts of (+)-sesamin and (+)-sesamolin in the sesame lines ITCFA2002 and #4294 during seed development. DAF; days after flowering. **c** The number of individual RILs (F6) generated by crossing #4294 and ITCFA2002 was plotted against (+)-sesamolin (%) relative to the total lignan content. red; RILs accumulating less than 5% (+)-sesamolin, blue; RILs accumulating 15–45% (+)-sesamolin. **d** Amounts (mg/g tissue) of (+)-sesamin, (+)-sesamolin and (+)-sesaminol triglucoside (STG) in mature seeds from (+)-sesamolin-deficient (red) and (+)-sesamolin-accumulating (blue) RILs. Values are mean ± SD. *$P < 0.05$, Mann–Whitney $U$-test, two-tailed. **e** (+)-sesamolin biosynthetic activity of the microsome fraction from sesame seeds. The enzyme assays were conducted with or without (+)-sesamin. Heat inactivation: incubation of the microsome fraction at 96 °C for 5 min prior to the assay. Values are mean ± SE ($n = 3$). *$P < 0.01$, Student's $t$-test

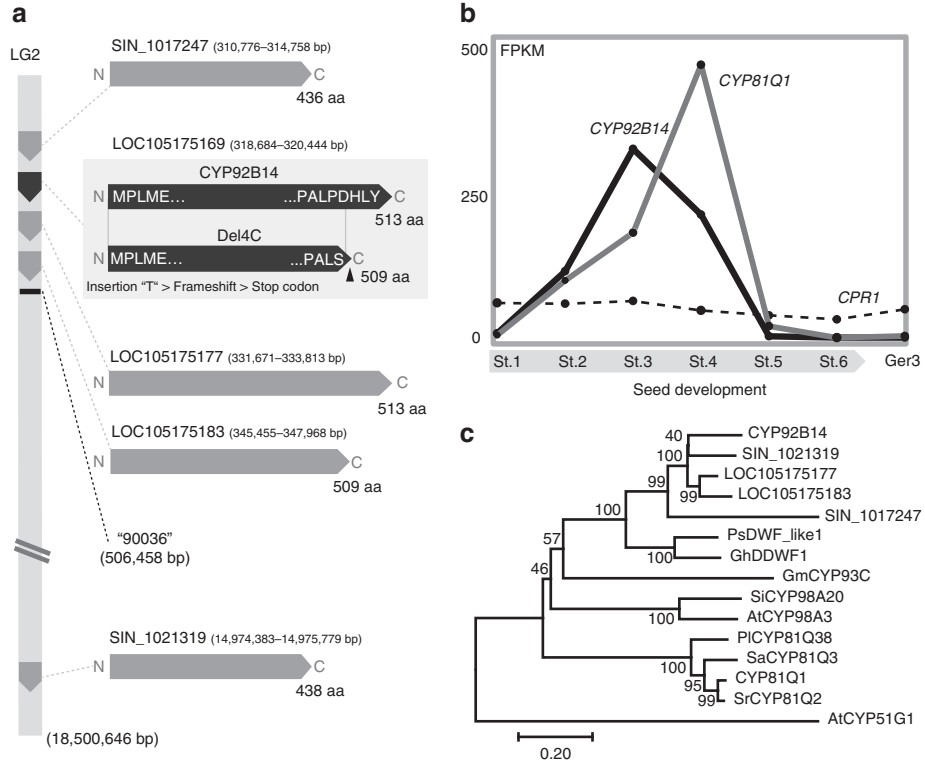

**Fig. 2** Genomic structure and expression of CYP92B14. **a** CYP92B14, LOC105175177, LOC105175183, and SIN_1017247 are the four P450s that constitute a gene cluster in LG2. SIN_1021319 is an additional P450 similar to CYP92B14. The arrowhead indicates the position of the single-nucleotide insertion. **b** Gene expression analysis obtained by RNA-Seq. FPKM; fragments per Kilobase Megareads. St.; seed maturation stage[10]. Ger3; 3 days after germination. **c** Phylogenetics of CYP92B14 determined by the neighbor-joining method using MEGA7. Bootstrap values (1000 replicates) are shown next to the branches. Scale bar represents the rate of substitution/site

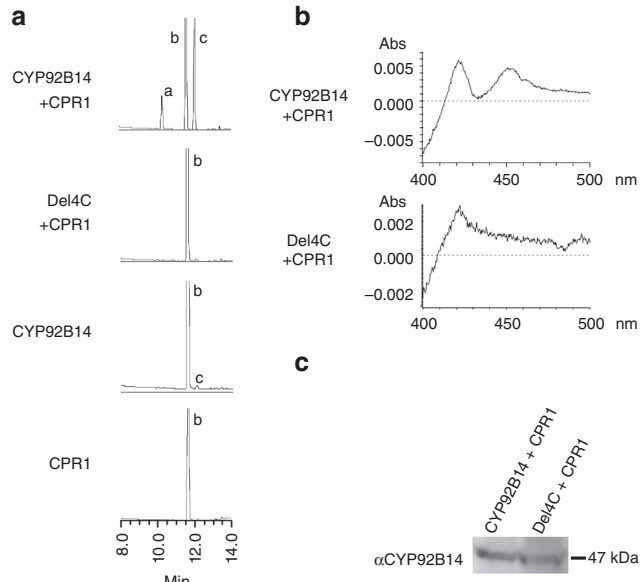

**Fig. 3** CYP92B14 biosynthesizes (+)-sesamolin and (+)-sesaminol from (+)-sesamin in vitro. **a** Microsome fractions prepared from a series of transformed yeasts were subjected to enzyme assays using (+)-sesamin as a substrate. a: (+)-sesaminol; b: (+)-sesamin; c: (+)-sesamolin. **b** CO-difference spectra of microsome fractions expressing *CYP92B14*. **c** Expression of recombinant CYP92B14 proteins in yeasts was immunologically detected with an anti-CYP92B14 antibody raised against the peptide sequence that is common to full-length CYP92B14 and Del4C

ITCFA2002 (Fig. 1b, Supplementary Table 1). The genetic analysis indicated that (+)-sesamolin was derived from (+)-sesamin, since (+)-sesamolin-deficient RILs accumulated increased amounts of (+)-sesamin compared to (+)-sesamolin-accumulating RILs (Fig. 1c, d). This notion was further supported by the higher level of (+)-sesamin in an another (+)-sesamolin-deficient cultivar Maruemon compared to its relative cultivars, Gomazou and Maruhime, both of which accumulate (+)-sesamolin[13]. The RILs were clearly grouped into two classes in a 1:1 ratio according to the level of (+)-sesamolin: of 160 lines, the relative content of (+)-sesamolin compared to total lignan was <5% in 81 lines but 15–45% in 79 lines (Fig. 1d, Supplementary Table 2). Moreover, the microsome fraction of the (+)-sesamolin-accumulating cultivar 'Masekin' exhibited biosynthetic activity of (+)-sesamolin from (+)-sesamin, but the activity in the fraction was abolished by heat inactivation prior to the assay (Fig. 1e). These collectively indicate that a single genetic locus encoding a microsomal enzyme is involved in the biosynthesis of (+)-sesamolin.

**CYP92B14 is responsible for (+)-sesamolin biosynthesis.** Restriction Site Associated DNA Sequence (RAD-Seq) analysis[14] of the segregated RIL lines and genetic mapping using 566 single-nucleotide polymorphism (SNP) markers covering over 90% of the RILs indicated that a marker, #90036, on a genomic contig, linkage group 2 (LG2)[15], was associated with (+)-sesamolin deficiency (Fig. 2a). There was a P450 gene cluster in the corresponding genomic region, where the open reading frame (ORF) of LOC105175169, designated as CYP92B14 by P450 mono-oxygenase nomenclature[16], harbored a thymine insertion shared in #4294 and Maruemon, generating a mutant lacking the four

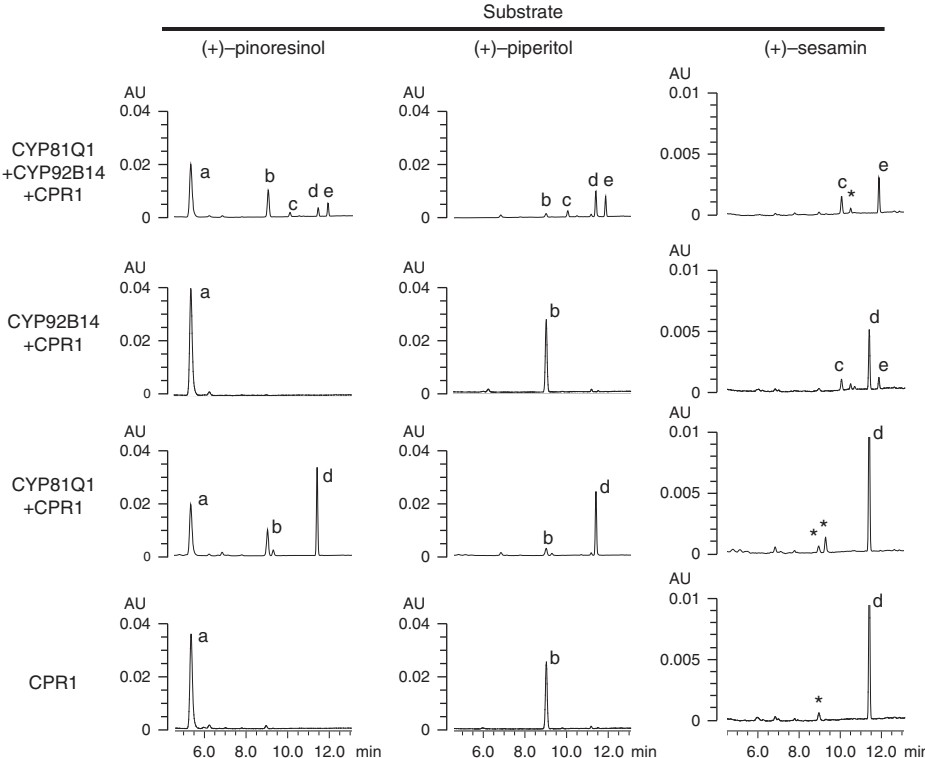

**Fig. 4** Co-expression of *CYP81Q1* ameliorates the activity of *CYP92B14* to produce (+)-sesamolin and (+)-sesaminol from (+)-sesamin in yeast cells. Yeast cell lines expressing either or both *CYP81Q1* and *CYP92B14* together with *CPR1* were subjected to bioassays using designated lignan as a substrate. Assay products after 48 h incubation were analyzed by HPLC at 283 nm. a: (+)-pinoresinol, b: (+)-piperitol, c: (+)-sesaminol, d: (+)-sesamin, e: (+)-sesamolin. Asterisks indicate unknown peaks

C-terminal amino acids (Del4C) (Fig. 2c, Supplementary Fig. 1). The expression of *CYP92B14* was increased during the mid-stage of seed development (Fig. 2b, Supplementary Figs. 2, 3). This profile coincided with that of sesamin synthase *CYP81Q1* (Fig. 1a)[10], and proceeded to the accumulation of (+)-sesamolin. In contrast, *S. indicum cytochrome P450 oxidoreductase 1 (CPR1)* was expressed throughout seed development. The expression level of *CYP92B14* was considerably higher than that of any other CYP gene except LOC105175177 (Supplementary Fig. 3). CYP92B14 and four other P450s identified in LG2 were grouped into the CYP92 family, which is widely found in gymnosperms and angiosperms. These five P450s are phylogenetically close to PsDWF_like1[17], which is involved in brassinosteroid biosynthesis, but distant from two phenylpropanoid-related P450s, coumarate 3-hydroxylase (C3H; CYP98A20) and CYP81Q1 (Fig. 2a, c, Supplementary Fig. 4).

**Biochemical characterization of CYP92B14.** Biochemical analysis showed that the microsome fraction from yeast cells expressing *CYP92B14* and *CPR1* converted (+)-sesamin to (+)-sesamolin (Fig. 3a). Furthermore, co-existence of CPR1 with CYP92B14 substantially enhanced the catalysis. Conversely, Del4C lost the ability to biosynthesize (+)-sesamolin (Fig. 3a). CO-difference spectrum analysis indicated that Del4C was expressed as an inactive form as shown by the absence of a peak at 450 nm (Fig. 3b), while its expression level in yeast was comparable to that of CYP92B14 (Fig. 3c, Supplementary Fig. 5). Moreover, *Del4C* in mature seeds of the (+)-sesamolin-deficient RIL (#119) was expressed at a level comparable to that of *CYP92B14* in the (+)-sesamolin-accumulating RIL (#89) (Supplementary Fig. 6). On the other hand, introduction of the amino acid substitution P509S to CYP92B14 did not result in the loss of

the catalytic activity (Supplementary Fig. 7). These data indicate that functional impairment originated from the lack of four amino acids in the C-terminus of CYP92B14, but not the lack of *CYP92B14* expression, is responsible for (+)-sesamolin deficiency in #4294. Intriguingly, CYP92B14 also generated (+)-sesaminol from (+)-sesamin, whereas Del4C lacked such activity (Fig. 3a), showing that CYP92B14 is a multifunctional enzyme that co-generates (+)-sesamolin and (+)-sesaminol.

**Effects of co-expression of CYP81Q1 on CYP92B14.** Various plant P450 enzymes have been implicated to interact and form complexes called metabolon with enzymes that catalyze sequential reactions in a metabolic pathway[18]. To assess whether CYP81Q1 affects the catalytic activity of CYP92B14, *CYP81Q1* was expressed in yeast cells together with *CYP92B14* and *CPR1*, and was subjected to the bioassays. Yeast cells co-expressing *CYP81Q1*, *CYP92B14*, and *CPR1* converted (+)-pinoresinol to (+)-sesaminol and (+)-sesamolin via (+)-sesamin, demonstrating sequential lignan biosynthesis by the distinct P450s (Fig. 4). Notably, when (+)-sesamin was supplied as a substrate, yeast cells harboring *CYP81Q1*, *CYP92B14*, and *CPR1* produced increased amount of (+)-sesaminol and (+)-sesamolin compared to those expressing *CYP92B14* and *CPR1* but without *CYP81Q1* (Fig. 4, Supplementary Fig. 7). Furthermore, while (+)-sesamin supplied as the substrate was detectable in yeast cells expressing *CYP92B14* and *CPR1*, it was completely consumed in yeast cells expressing *CYP92B14* and *CYP81Q1* together with *CPR1*. On the other hand, CYP81Q1 alone did not react with (+)-sesamin, and CYP92B14 alone showed no apparent catalytic activity toward (+)-pinoresinol and (+)-piperitol. These data suggest catalytic cooperation between CYP81Q1 and CYP92B14 for biosynthesis of (+)-sesamolin and (+)-sesaminol.

**CYP92B14 mediates unique oxygenation schemes.** The oxygenation scheme by CYP92B14 was expected to be unique, since CYP92B14 co-generated twin products, (+)-sesaminol and (+)-sesamolin, from a single substrate (+)-sesamin. CYP92B14 incorporated oxygen from molecular oxygen, but not $H_2O$, for the oxygenation catalysis as with other P450s (Supplementary Fig. 8). NMR analysis of CYP92B14 assay products showed that the deuterium-labeling content in (+)-sesaminol and

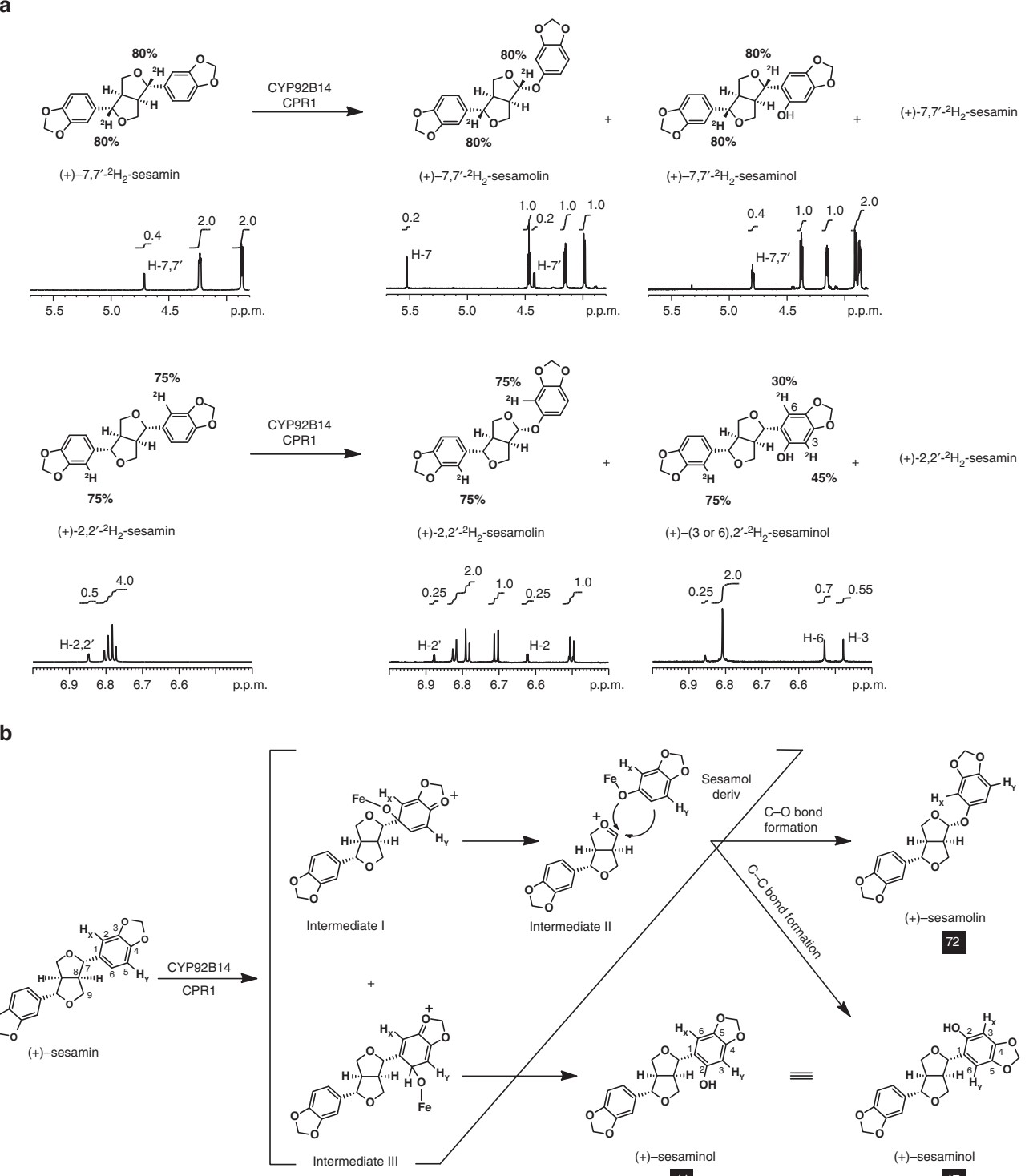

**Fig. 5** CYP92B14 generates (+)-sesamolin and (+)-sesaminol from (+)-sesamin through multiple oxygenation schemes. **a** Enzyme assays using deuterated (+)-sesamin revealed the sites of initial oxidation by CYP92B14. No differences in the positions and deuterium-labeling content in (+)-7,7'-$^2H_2$-sesamin and (+)-2,2'-$^2H_2$-sesamin were observed between before and after the reaction. Values (%) represent deuterium-labeling content calculated from NMR data. Note that the C3 and C6 positions of (+)-sesaminol correspond to the C5 and C2 positions of (+)-sesamin, respectively. **b** Proposed reaction pathways from (+)-sesamin to (+)-sesamolin and (+)-sesaminol. White letters indicate the molar ratio of respective enzyme assay products deduced from NMR analysis. The total amount of produced (+)-sesamolin and (+)-sesaminol was set as 100

(+)-sesamolin was essentially identical to that in the substrate (+)-7,7′-$^2$H$_2$-sesamin (80% $^2$H for each position) (Compound **1**) (Fig. 5a). Alternatively, deuterium-labeling content in (+)-sesaminol was changed in the C3 and C6 positions (C3: 45% $^2$H, C6: 30% $^2$H) when (+)−2,2′-$^2$H$_2$-sesamin (75% $^2$H for each position) (Compound **2**) was used as a substrate (Fig. 5a). The deuterium-labeling content at the C2 or C2′ positions of (+)-sesamolin was unchanged from that of the substrate (+)-2,2′-$^2$H$_2$-sesamin. These results show that, for the biosynthesis of (+)-sesamolin, CYP92B14 catalyzes C–O bond formation between the intermediate II and sesamol via intermediate I following the initial oxidation of (+)-sesamin at C1 position and the continuous cleavage of the furan and aromatic rings (Fig. 5b), which is comparable to that of P450-dependent C–C bond cleavage of *p*-hydroxybenzyl alcohol[19]. On the other hand, CYP92B14 biosynthesizes (+)-sesaminol by two separate pathways: (i) C–C bond formation between the intermediate II and sesamol as in the case of sesamolin biosynthesis, and (ii) direct oxygenation of the aromatic ring via intermediate III following the initial oxidation of (+)-sesamin at the C6 position (Fig. 5b). These oxygenation positions catalyzed by CYP92B14 for (+)-sesamolin and (+)-sesaminol biosynthesis were clearly distinct from that of (+)-episesaminone, which is tautomerically generated by oxygenation at the C7 position through hemiacetal intermediate[3,20]. NMR analysis of CYP92B14 assay products using (+)−2,2′-$^2$H$_2$-sesamin and (+)−7,7′-$^2$H$_2$-sesamin also showed that CYP92B14-mediated catalysis yielded (+)-sesamolin through ORA, (+)-sesaminol through ORA and (+)-sesaminol through direct oxygenation in a molar ratio of 72:17:11, respectively, (Fig. 5b).

## Discussion

CYP92B14 catalyzed the intricate reactions; a novel oxygenation designated as ORA and a well-characterized direct oxygenation of aromatic ring. ORA involved (i) the oxygenation of carbon atom on the aromatic ring system substituted with alkyl group, possibly via cation intermediate, (ii) the cleavage of the C–C bond between the aromatic and α-oxy-substituted alkyl groups, followed by (iii) the addition of the generated phenol adduct and the oxonium intermediate (Fig. 5b). Enzyme assays using deuterated (+)-sesamin clearly indicated that CYP92B14 catalyzed the oxidation of aromatic ring of (+)-sesamin. P450 monooxygenases in general operate the initial oxidation of a phenolic substrate through electrophilic attack of FeO complex to the aromatic ring, which generates a cation rather than radical intermediate[21]. Therefore, CYP92B14 likely generates cation intermediates I and III in the initial oxidation of (+)-sesamin. The preference of the generation of intermediate I over intermediate III, and that of C–O over C–C bonding in the addition of the phenol adduct to the oxonium intermediate (Fig. 5b) would largely reflect the structural constraints of the active site. Although plant specialized metabolites rarely harbor an insertion of an oxygen atom between aromatic and α-oxy-substituted alkyl group as in (+)-sesamolin[11], the initial oxygenation step of ORA might be widespread outside sesame lignans. For example, a quinoid glycoside forsythenside B[22] from *Forsythia suspensa*, a pretocarpan hydroxycristacarpone[23] from *Erythrina orientalis* and a sesquiterpene hydroxycacalolide[24] from *Ligularia virgaurea* all possess 4-alkyl-4-hydroxy-2,5-cyclohexadienone partial structures generated by the oxidation of 4-alkylphenol compounds, respectively (Supplementary Fig. 9). Moreover, a characteristic hydroxyl group at the C1-position of the B-ring in protoapigenone and protogenkwanone in the fern genera *Thelypteris* and *Pseudophegopteris* likely to be generated by oxygenation analogous to that by CYP92B14 (Supplementary Fig. 9)[25,26]. Furthermore, it is plausible that 5,7-dihydroxychromone accumulated in *Magnifera*

*foetida* and various other plant species is derived from C–C bond cleavage of the flavanone naringenin (Supplementary Fig. 9)[27–32]. Since no enzymes that catalyze these oxygenation reactions have been identified so far, CYP92B14 is the first example of enzymes that mediates the oxygenation of the carbon atom on the aromatic ring system substituted with alkyl group. It is of particular interest whether the putative P450 enzyme encoded by LOC105175177 exhibits any enzyme activity toward sesame lignans. However, our attempts to identify the activity of the P450 enzyme have not been successful so far, since we failed to express LOC105175177 in yeast (Supplementary Fig. 5). Future establishment of the expression system of LOC105175177 and its functional characterization in comparison to CYP92B14 may provide further insights into the molecular mechanism of ORA that is mediated by this type of P450 enzymes. ORA might also be applicable to other classes of specialized metabolites including isoquinoline alkaloids and lignin polymers, since these metabolites have α-oxy-substituted aromatic ring as a common structural feature (Supplementary Fig. 9). In particular, typical lignin polymers contain multiple putative sites for oxygen insertion by ORA, implicating that the use of enzymes with catalytic activity similar to CYP92B14 facilitates full decomposition of lignan units that are otherwise indigestible by other classes of lignin-degrading enzymes[33,34].

Although the C-terminal region of P450s is unlikely to be directly associated with the substrate-binding domain[35], the C-terminal region seems to be indispensable for the catalytic activity of P450. For example, an amino acid substitution P497R in *Homo sapiens* CYP27B1 leads to the functional impairment and is responsible for vitamin D-dependent rickets type I and cerebrotendinous xanthomatosis[36]. On the other hand, in the case of CYP92B14 mutant Del4C, which lacks the last four residues in its C-terminus and contains P509S substitution because of the thymine insertion, is unable to catalyze the reaction (Fig. 2a, Supplementary Fig. 1), whereas P509S mutant harboring full-length sequence remains active (Supplementary Fig. 7). Although the molecular mechanism behind the loss of oxygenation activity of these P450s is unclear, these data corroborate the implicit functional importance of the C-terminal region of P450s[36].

Importantly, sesamol and (+)-samin, a hydrate of the oxonium intermediate, were generated when CYP92B14 was supplied with (+)-sesamin (Supplementary Fig. 10). Moreover, the supplementation of these two compounds clearly inhibited the biosynthetic activity of CYP92B14 to form (+)-sesamolin and (+)-sesaminol from (+)-sesamin (Supplementary Fig. 11). These results supported the notion that sesamol and intermediate II were potent reaction intermediates of (+)-sesamolin and (+)-sesaminol biosynthesis (Fig. 5b), and partly explained why the total amount of (+)-sesamolin and (+)-sesaminol produced was substantially smaller than the amount of (+)-sesamin consumed in the yeast bioassay (Fig. 4, Supplementary Fig. 10). The lack of sesamol and (+)-samin production by Del4C (Supplementary Fig. 10)[37] indicated that Del4C was unable to cleave the inert C–C bond between the furan and aromatic rings of (+)-sesamin.

The identification of CYP92B14 uncovered the final steps in the biosynthetic pathway of all major lignans in *S. indicum*, (+)-sesamin, (+)-sesaminol and (+)-sesamolin, being (+)-sesamin as a precursor for (+)-sesaminol and (+)-sesamolin (Supplementary Fig. 12). This was biochemically confirmed by the result that the expression of *CYP81Q1*, *CYP92B14*, and *CPR1* in yeast cells was sufficient to produce (+)-sesaminol and (+)-sesamolin from (+)-pinoresinol (Fig. 4). (+)-Sesamolinol, although previously proposed to be a putative precursor for (+)-sesamolin biosynthesis[38], is unlikely to be directly related to the biosynthesis of (+)-sesamolin, since (+)-sesamolinol and (+)-piperitol were not accepted as substrates by CYP81Q1 and CYP92B14,

respectively (Fig. 4)[10,39]. However, there might be additional enzymes, other than CYP92B14, that are responsible for (+)-sesaminol biosynthesis in *S. indicum*, since sesamolin-deficient RILs accumulate (+)-sesaminol triglucoside (STG) in vivo to a level comparable to that of sesamolin-producing RILs, while Del4C does not biosynthesize (+)-sesaminol in vitro (Figs. 1d, 3a).

Co-generation of structurally different metabolites through ORA, as evidenced in this study, reveals the outstanding enzymatic plasticity of P450s to elaborate structurally diverged specialized metabolites for increased environmental fitness. CYP92B14 could have recently evolved following the establishment of (+)-sesamin biosynthesis in an ancestor of *Sesamum* spp., given that (i) (+)-sesamolin is a lineage-specific lignan in *Sesamum* spp.[40], (ii) (+)-sesamin is the specific substrate for CYP92B14, and (iii) CYP81Q1 and CYP92B14 are phylogenetically distinct P450s despite the high structural similarities between their substrates/products. The capability of CYP81Q1 to ameliorate the activity of CYP92B14 (Fig. 4) further substantiated the versatility of P450s in producing multiple specialized metabolites. Enzymes that were involved in primary metabolism (i.e., glycolysis, amino acid biosynthesis and TCA cycle), as well as various specialized metabolism in plants, in particular, have been implicated to form multiple enzyme complexes that carry out sequential enzymatic reactions[18,41]. However, major effort has been made to detect physical association of enzymes that form metabolons, and only few examples biochemically rationalized the formation of such complexes[18,42–44]. Therefore, our data suggesting that the activity of CYP92B14 is functionally associated with CYP81Q1 may provide a novel system to study structure-function relationship of enzymes that function cooperatively.

Consequently, our data not only complete the biosynthesis of major lignans in sesame, but provide tools for heterologous expression of valuable dietary lignans. Furthermore, our findings open a gate for novel chemical perspective with respect to oxygenation through ORA in various classes of specialized metabolites that possess heteroatoms at the α-position beyond sesame lignans (Supplementary Fig. 9), thus established a novel molecular basis for enlarging the structural diversity of specialized metabolites in nature.

## Methods

**Plant materials.** *Sesamum indicum* cv. Masekin plants were cultivated in a field in Osaka, Japan. Immature seeds were collected along the developmental stage index as described previously[10]. *S. indicum* cv. Maruemon was a gift from National Agriculture and Food Research Organization (NARO). *S. indicum* ITCFA2002 was kindly provided by T. Amano (ITOCHU) and T. Kondo (Nippon Norin Seed). F1 plants and RILs from a cross between #4294 and ITCFA2002 were grown in a greenhouse at University of Toyama.

**RAD-Seq and genetic analyses.** Genomic DNA of RILs and their parental lines (#4294 and ITCFA2002) were isolated from young leaves using the cetyl trimethylammonium bromide (CTAB) method with some modifications. Briefly, a fine frozen tissue powder was washed three times with washing buffer (0.5 M HEPES (pH 8.0), 0.1% polyvinylpyrrolidone, 4% β-mercaptoethanol) and then subjected to DNA extraction by the CTAB method, but omitting the CsCl precipitation step. The genomic DNAs were adjusted to 50 ng/μl and used for RAD-Seq analysis as described previously[45]. RIL SNP was genotyped by the derived Cleaved Amplified Polymorphic Sequence (dCAPS) method[46] using the primer sets 5′-GCCATCTTTGGTTTCTGGTG-3′ and 5′-TGGGACTGACTCAACGGC-3′ for the #90032 marker, and restriction enzyme *Hae* III. PCR was performed using TAKARA Ex Taq Hot Start Version (TAKARA) DNA polymerase under the following conditions: an initial cycle at 94 °C for 2 min, 35 cycles at 98 °C for 10 s, 60 °C for 30 s, 72 °C for 30 s, and a final extension step at 72 °C for 2 min. Linkage analysis of RAD markers and the (+)-sesamolin content trait was performed using Antmap ver. 1.2 software[47].

**RT-PCR.** Total RNA was extracted from developing seeds using RNeasy Plant Mini Kit (Qiagen). After DNase I (TAKARA) treatment, cDNA was synthesized from oligo dT primer using PrimeScript RT reagent Kit (TAKARA). Reactions were conducted according to manufactures' instructions. PCR was performed using primer sets (Supplementary Table 3) under the conditions described above. Primers for *UBQ6* were designed as reported previously[48].

**Phylogenetic analysis.** Phylogenetic analysis of CYP92B14 and 12 other P450 monooxygenases from plants was conducted using the UPGMA method with MEGA7[49]. The accession numbers of the genes subjected to phylogenetic analysis are as follows; SiCYP92B14, LC199944; SiCYP98A20, NP_001291339; SiCYP81Q1, BAE48234; SrCYP81Q2, BAE48235; SaCYP81Q3, BAE48236; PlCYP81Q38, BAP46307; GmCYP93C, SBU44858; AtCYP51G1, NP_172633; AtCYP98A3, OAP09214; PsDWF_like1, AAG44132; and GhDDWF1, ABA01477.

**Chemicals.** (+)-Pinoresinol and (+)-sesaminol (Nagara Science) were used without further purification. (+)-Sesamin (ChromaDex) was sequentially purified by normal, followed by reverse phase column chromatography. (+)-Sesamolin was isolated during the purification of (+)-sesamin. For the synthesis of (+)-samin, (+)-sesamolin was treated with 3N aqueous hydrogen chloride in dioxane at 45 °C for 1 h[37]. For the synthesis of (+)-piperitol, (+)-sesamin was treated with diisobutyl aluminum hydride in toluene at 140 °C for 1 h[39]. Sesamol and other chemical reagents, including solvents, were purchased from Tokyo Chemical Industry, Sigma-Aldrich or Nacalai Tesque.

**Chemical synthesis of stable isotope-labeled sesamin.** (+)-7,7′-$^2$H$_2$-sesamin (**1**): (+)-Sesamin (270 mg, 0.762 mmol) was subjected to 4.0 kgf/cm$^2$ of hydrogen in the presence of Pd(en) (270 mg) in a mixture of 1,4-dioxane (4.2 ml) and deuterium oxide (1.8 ml) at 110 °C for 3 days[50]. After filtration of the reaction mixture, the filtrate was purified by reverse phase column chromatography to provide deuterated **1** (153 mg, 0.429 mmol, 56%). The $^2$H content at the C7 and C7′ of deuterated (+)-sesamin was 80%, as estimated by comparison of the hydrogen intensity of H-7, 7′ and H-9a, 9′a in a $^1$H NMR experiment.

**1**: mp 123–124 °C; [α]$_D^{25}$ +67.7° (c 0.16, CHCl$_3$); $^1$H NMR (800 MHz, CDCl$_3$) δ3.05 (2 H, m, H-8, 8′), 3.87 (2 H, dd, *J* = 2.4, 9.2 Hz, H-9a, 9′a), 4.23 (2 H, dd, *J* = 6.6, 9.2 Hz, H-9b, 9′b), 4.71 (2 H, d, *J* = 3.5 Hz, H-7, 7′), 5.95 (4 H, s, H-10, 10′), 6.78 (2 H, d, *J* = 8.0 Hz, H-5, 5′), 6.80 (2 H, br d, *J* = 8.0 Hz, H-6, 6′), 6.85 (2 H, br s, H-2, 2′); $^{13}$C NMR (200 MHz, CDCl$_3$) δ 54.2, 54.3 (C-8, 8′), 71.66, 71.67 (C-9, 9′), 85.4 (t, *J* = 22.0 Hz, C-7, 7′), 85.8 (C-7, 7′), 101.1 (C-10, 10′), 106.5 (C-2, 2′),108.2 (C-5, 5′), 119.3, 119.4 (C-6, 6′), 134.97, 135.03 (C-1, 1′), 147.0, 147.1 (C-4, 4′), 148.0 (C-3, 3′). HRMS (ESI): [M+Na]$^+$ calculated for C$_{20}$H$_{16}$D$_2$O$_6$Na: *m/z* 379.3549, found: 379.3542. NMR spectra of **1** are shown in Supplementary Figs. 13–15.

(+)-2,2′-$^2$H$_2$-sesamin (**2**): Synthesis of (+)-2,2′-$^2$H$_2$-sesamin was performed basically as reported previously[51]. To a solution of (+)-sesamin (354 mg, 1.0 mmol) in dry tetrahydrofuran (5 ml) was added n-BuLi (1.5 ml, 1.6 mol/L in toluene) at −78 °C under argon. After stirring for 30 min, methanol-d$_4$ (0.5 ml) was added, the solution was allowed to warm to room temperature, then the solution was partitioned between H$_2$O and ethyl acetate, the organic layer washed with saturated NaCl solution, dried (MgSO$_4$) and concentrated in vacuo to dryness. The resulting residue was purified by reverse phase column chromatography to provide deuterated **2** (185 mg, 0.519 mmol, 52%). The $^2$H content at the C2 and C2′ of deuterated (+)-sesamin was 75%, as estimated by comparison of the hydrogen intensity of H-2, 2′ and H-9a, 9′a in a $^1$H NMR experiment.

**2**: mp 123–124 °C; [α]$_D^{25}$ +68.6° (c 0.18, CHCl$_3$); $^1$H NMR (800 MHz, CDCl$_3$) δ3.05 (2 H, m, H-8, 8′), 3.87 (2 H, dd, *J* = 3.7, 9.2 Hz, H-9a, 9′a), 4.23 (2 H, dd, *J* = 6.7, 9.2 Hz, H-9b, 9′b), 4.71 (2 H, d, *J* = 4.2 Hz, H-7, 7′), 5.95 (4 H, s, H-10, 10′), 6.78 (2 H, d, *J* = 8.0 Hz, H-5, 5′), 6.80 (2 H, br d, *J* = 8.0 Hz, H-6, 6′), 6.85 (2 H, br d, *J* = 1.6 Hz, H-2, 2′); $^{13}$C NMR (200 MHz, CDCl$_3$) δ54.3 (C-8, 8′), 71.6 (C-9, 9′), 85.70, 85.73 (C-7, 7′), 101.0 (C-10, 10′), 106.2 (t, *J* = 24.8 Hz, C-2, 2′), 106.4 (C-2, 2′), 108.1 (C-5, 5′), 119.3 (C-6, 6′), 134.9, 135.0 (C-1, 1′), 147.05, 147.06 (C-4, 4′), 147.85, 147.91 (C-3, 3′); HRMS (ESI): [M+Na]$^+$ calculated for C$_{20}$H$_{16}$D$_2$O$_6$Na: *m/z* 379.3549, found: 379.3540. NMR spectra of **2** are shown in Supplementary Figs. 16–18.

**LC–MS analysis of stable isotope labeled compounds.** High-resolution mass spectra of all products including deuterated or $^{18}$O-labeled compounds were obtained using an ion-trap time-of-flight mass spectrometer (Shimadzu LCMS-IT-TOF). A YMC-Triart C18 (100 × 2.1 mm, 3.0 μm) column was used for separation of enzymatic products derived from (+)-sesamin or deuterated (+)-sesamin. The mobile phase consisted of 0.1% formic acid in MilliQ water (A) and 0.1% formic acid in methanol (B) and used for a linear gradient elution (0–7–10–10.01–15 min, 30–90–90–30–30%B). Positive and negative ions were measured simultaneously. The parameters were as follows: nebulizer gas, 1.5 L/min; drying gas pressure, 190 kPa; CDL temperature, 200 °C; block heater temperature, 200 °C; detector, 1.64 kV; and interface, 4.5 kV (positive mode) and −3.5 kV (negative mode).

**NMR analysis of stable isotope labeled compounds.** All NMR spectra of all products including deuterated compounds were acquired on a Bruker AVANCE III HD 800 spectrometer equipped with a 5-mm TCI cryogenic probe and Z-axis gradient (Bruker Biospin). All spectra were measured at 25 °C, using 5 mm NMR tubes. Standard Bruker pulse sequences were employed. The data analyses were

carried out with Bruker TopSpin 3.2 software (Bruker Biospin). The acquisition parameters were as follows: [1]H NMR: number of data points 64 K; spectral width 16025 Hz; relaxation delay 10 s; number of scans 64; receiver gain 3.2; [13]C NMR: number of data points 64 K; spectral width 48077 Hz; relaxation delay 2 s; number of scans 1024; receiver gain 203; [2]H NMR: number of data points 64 K; spectral width 4092 Hz; relaxation delay 1 s; number of scans 1024; receiver gain 128. Chemical shift was reported in parts per million (p.p.m.) relative to the peaks from chloroform-$d$ (CDCl$_3$, $\delta_H$ 7.26 p.p.m., $\delta_C$ 77.0 p.p.m.). The data for [1]H NMR are reported as follows: chemical shift, multiplicity (s = singlet, d = doublet, t = triplet, q = quartet, m = multiplet), integration and coupling constants.

**RNA-seq.** Total RNA was extracted from six developmental stages of seeds and germinating seeds (3 days after imbibition) using RNeasy Plant Kit (Qiagen). The developmental stages of sesame seeds were described previously[10]. The RNA samples were treated with DNase Set (Qiagen) to remove contaminating genomic DNA. The quality of each RNA sample was evaluated using BioAnalyzer (Agilent Technologies) with RNA6000 Nano Chip. A 1 mg aliquot of total RNA from each sample was used to construct cDNA libraries using TruSeq Stranded Total RNA with Ribo-Zero Plant (Illumina) according to the manufacturer's instructions. The resulting cDNA library was validated using BioAnalyzer with DNA1000 Chip and quantified using Cycleave PCR Quantification Kit (TAKARA). Paired-end sequencing using 2 × 101 cycles was performed using HiSeq 1500 sequencing system (Illumina) in the high output mode.

Total reads were extracted using CASAVA v1.8.2 (Illumina), then PCR duplicates, adaptor sequences, and low quality reads were removed from the extracted reads as follows. Briefly, if the first 10 bases of two reads were identical and the entire reads showed >90% similarity, the reads were considered to be PCR duplicates. Base calling from the 5′ to the 3′ end was stopped when the frequency of accurately called bases dropped to 0.5. The remaining reads were assembled using Trinity (25 February 2013 release) by normalization with maximum coverage to 30. For each sample, the Fragments per Kilobase Megareads (FPKM) values were calculated for contigs containing cDNA sequences responsible for CYP92B14, CYP81Q1, and CPR1 using Bowtie.

**qPCR analysis.** qPCR analysis was performed as described previously[52]. Briefly, 18S rRNA was used as a reference gene for normalization and presented as expression ratios relative to control via the ΔΔCt method. Oligonucleotides used for the analysis are listed in Supplementary Table 3.

**Enzyme assays using sesame microsome fractions.** Fresh immature seeds (1 g) in Stage 4 were frozen in liquid nitrogen and homogenized to a fine powder under liquid nitrogen cooling using TissueLyzer II (Qiagen), then resuspended in 2 ml of extraction buffer I containing 100 mM Tris-Cl (pH 8.0), 0.1% β-mercaptoethanol, 1× Complete EDTA-free protease inhibitor (Roche) and 20% glycerol. The homogenate was desalted using a MicroSpin HR S-400 column (GE Healthcare) and centrifuged at 21,000×$g$ for 10 min at 4 °C, then the supernatant was ultracentrifuged at 70,000×$g$ for 1 h at 4 °C. The pellet was resuspended in 200 μl of Extraction buffer I. The microsome fraction (10 μl) was mixed with 100 μM (+)-sesamin and 1 mM NADPH in a final volume of 50 μl and incubated for 30 min at 37 °C. Heat inactivation was conducted by incubating the microsome fraction at 96 °C for 5 min prior to the enzyme assay. The reaction was terminated by the addition of an equal volume of acetonitrile, the reaction mixture was filtered through a Millex-LH syringe filter (Merck Millipore), then analyzed by using an HPLC equipped with a photodiode array detector (Waters)[53] or an LC–MS equipped with a fluorescent detector (Shimadzu), unless otherwise stated.

**Plasmid construction and yeast transformation.** cDNAs of CYP92B14, Del4C, CPR1, and CYP81Q1 were amplified with cDNAs derived from sesame seeds at developmental stage 4 as the PCR templates and specific primer sets, described in Supplementary Table 3. Amplified fragments were inserted into the yeast expression vectors pYE22m, pDEST52 (Invitrogen), and pJHXSBP-His using standard procedures. The plasmid list and vector information are shown in Supplementary Table 3. Individual genes cloned in each vector were verified by DNA sequencing of both strands. Yeast INVsc strain (Invitrogen) was transformed using a conventional method with yeast expression vectors carrying *Sesamum* genes. The transformed yeast colonies were isolated on SD agar plates without selection compounds for 2 days at 30 °C.

**Enzyme assays using yeast microsome fractions.** Yeast cell lines expressing CYP92B14 and CPR1, Del4C and CPR1, CYP92B14 alone or CPR1 alone were precultured in 50 ml SD medium lacking respective amino acids overnight at 120 r. p.m. 30 °C. The saturated culture was transferred to 950 ml SD liquid medium without respective amino acids and cultured at 120 r.p.m. 30 °C until the OD$_{600}$ reached 2.0. The cells were harvested by centrifugation at 10,000×$g$ for 15 min at 4 °C. The pellet was rinsed once and resuspended in 10 ml of Extraction buffer I, followed by sonication at 4 °C. The homogenate was centrifuged at 21,000×$g$ for 10 min at 4 °C, then the supernatant was ultracentrifuged at 70,000×$g$ for 1 h at 4 °C. The pellet was resuspended in 600 μl of extraction buffer I and subjected to enzyme assays. The microsome fraction (50 μl) was mixed with 100 μM (+)-sesamin and 1

mM NADPH in a final volume of 100 μl and incubated for 60 min at 37 °C. The reaction was terminated by the addition of an equal volume of acetonitrile, and the mixture was filtered through a Millex-LH syringe filter (Merck Millipore), then subjected to HPLC analysis.

**Bioassay experiments in yeast cells.** For 1 ml-scale experiments, single colonies of yeast strains expressing the ORF sets CYP92B14+CPR1 or Del4C+CPR1 were cultured in 3 ml synthetic defined liquid medium overnight at 30 °C with rotary shaking at 120 r.p.m. The stationary phase culture (50 μl) was transferred to 1 ml of fresh medium in 24-well plates supplemented with 100 μM of the lignan being tested as a substrate in the experiment. The cultures were further incubated for 24 h unless otherwise stated at 30 °C with rotary shaking at 120 r.p.m. The efficiency of lignan extraction from the culture by the addition of an equal volume of acetonitrile was verified (Supplementary Fig. 19). Briefly, the cells were collected together with the medium and disrupted by sonication. The homogenate (50 μl) was mixed with 50 μl acetonitrile and centrifuged at 21,000×$g$ for 10 min. The supernatant was filtered through a Millex-LH syringe filter and subjected to LC–MS analysis. The identity of the reaction products was confirmed by comparing the retention time and parent mass with authentic lignan standards.

For the [2]H-labeling studies, yeast cells carrying CYP92B14+CPR1 were cultured in 600 ml of liquid medium containing 200 μM of [2]H-labeled sesamin for 48 h at 30 °C with rotary shaking at 120 r.p.m., then extracted twice with an equal volume of ethyl acetate. The organic layer was dehydrated with magnesium sulfate, dried in vacuo to dryness, and reconstituted with 400 μl of 50% acetonitrile. The extract was fractionated by HPLC, and the fractions containing enzyme reaction products were subjected to MS and NMR analyses. NMR spectra of enzyme reaction products are shown in Supplementary Figs. 20–23.

For the [18]O-labeling studies, either yeast medium prepared with [18]O-labeled H$_2$O (Taiyo Nippon Sanso), or yeast medium prepared with non-labeled H$_2$O and aerated with [18]O-labeled O$_2$ (GL Science), was used for the yeast bioassay experiments. Briefly, the cells were cultured in 1 ml of [18]O-labeled medium containing 100 μM sesamin in 24-well plates for 48 h at 30 °C with rotary shaking at 120 r.p.m. Alternatively, 50 μl of stationary phase liquid cultures of yeast cells were sub-cultured in 1 ml of non-labeled liquid medium bubbled in advance with [18]O-labeled O$_2$ at a rate of 1 ml/min for 10 min. The air space in the test tubes was replaced with [18]O-labeled O$_2$, then the tubes were sealed tightly and the yeast cells were cultured for 4 h at 30 °C with rotary shaking at 120 r.p.m. The reactions were terminated by the addition of an equal volume of acetonitrile, filtered through a Millex-LH filter, and analyzed by LC–MS.

**CO-difference spectroscopy.** Yeast cells expressing the ORF sets CYP92B14 +CPR1 or Del4C+CPR1 were cultured in two three liter flasks containing one liter medium to an OD$_{600}$ of 2.0. The cells were collected by centrifugation at 10,000×$g$ for 15 min at 4 °C and rinsed once with 50 ml of distilled water. The cells were resuspended in extraction buffer I, disrupted by sonication on ice, then the homogenate was centrifuged at 21,000×$g$ for 15 min at 4 °C, followed by ultracentrifugation at 75,000×$g$ for 1 h at 4 °C. The pellet was resuspended in 600 μl of extraction buffer. CO-difference spectroscopy was conducted basically as described previously[54]. Briefly, the absorbance spectrum (400–500 nm) of the obtained microsome fraction was measured, then the microsome fraction was reduced by the addition of sodium dithionite and the spectrum of the reduced form was measured. The microsome fraction was then saturated with CO by bubbling CO for 1 min and the spectrum of the CO-bound form of P450 monooxygenase was measured.

**Immunoblot analysis.** Microsomal proteins of yeast cells expressing CPR1 with either CYP92B14 or Del4C were separated by SDS-PAGE and subjected to immunoblot analysis. Briefly, 20 μg of microsomal proteins per lane was separated by SDS-PAGE using a 5–20% acrylamide gradient gel and blotted onto a Hybond-P VDF membrane (GE Healthcare). Primary antibody reactions used 500× diluted rabbit polyclonal antibodies raised against a 1:1 mixture of the oligopeptides NH$_2$-C+WRQARKIYLSEVF-COOH and NH$_2$-C+QFLRLHDKVFASRP-COOH (Eurofins Genomics), both of which are perfectly conserved among all the P450 monooxygenases analyzed in the immunoblot analysis. Secondary antibody reactions used 5000× diluted rabbit IgG-HRP (GE Healthcare). Chemiluminescent detection was performed using ECL Select (GE Healthcare) as a substrate with Amersham Imager 600 (GE Healthcare) (Supplementary Fig. 5).

**Data availability.** The authors declare that fastq files for RNASeq data have been deposited in the NCBI SRA repository under Accession ID PRJNA350858. Sequences for CYP92B14 and CPR1 have been deposited in GenBank under accession codes LC199944 and LC209223, respectively. The authors declare that all other data supporting the findings of this study are available within the manuscript and its Supplementary Files or are available from the corresponding author upon request.

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

## Acknowledgments

We thank T. Amano and T. Kondo for ITCFA2002 seeds, and NARO for Maruemon seeds. We are also grateful to K. Namba, T. Sakaki, T. Umezawa, T. Waki, S. Takahashi, T. Nakayama, H. Takagi, H. Satake, K. Shimamoto, S. Nakanishi, H. Hatanaka, M. Fukuchi-Mizutani, M. Nakai, Y. Ono, and K. Yamada for fruitful discussion. This research was supported partly by JST PRESTO JP16H01473 (A.J.N.) and Grants-in-Aid for Scientific Research (C) JP25450003 (M.P.Y.).

## Author contributions

M.P.Y. and M.H. designed the research, S.Y., T.W., and M.P.Y. performed genetic analyses, E.O. and H.T. cloned CYP92B14, A.J.N. conducted RAD-Seq analyses, E.O. and A.S. analyzed RNA-Seq data, J.M. and M.H. performed biochemical analyses, M.N. and M.M. performed CO-difference spectra analysis, M.H., M.T., and T.A. conducted chemical syntheses, M.H. and S.M. performed NMR analyses, and J.M., E.O., M.P.Y., and M.H. wrote the manuscript. J.M., E.O., and S.Y. contributed equally to this work.

## Additional information

**Competing interests:** E.O. and H.T. are employees of Suntory Global Innovation Center, Ltd. Suntory Foundation for Life Sciences is a non-profit organization. The corresponding authors had full access to all the data in the study and had final responsibility for the decision to submit for publication. All other authors declare no competing financial interests.

