## [Peer Review File · Nature Communications]

Reviewers' comments:

Reviewer #1 (Remarks to the Author):

Sesame seeds contain (+)-sesamin, (+)-sesamol and (+)-sesaminol as the major lignans, which are interesting constituents due to their health-promoting activities. Here, the authors identify a new cytochrome P450 enzyme (CYP92B14), which converts (+)-sesamin to (+)-sesamol and (+)-sesaminol. CYP92B14 expression correlated with that of CYP81Q1 (sesamin synthase). A unique oxygenation scheme is proposed, based on the deuterium-labelling content of the products, as determined by NMR. CYP92B14 catalyses both direct oxygenation and oxidative rearrangement. The initial oxygenation step of α -oxy-substituted aryl groups may be widespread outside sesame lignans. The results reported are novel and interesting for a wider field and they will stimulate further research into this area.

(+)-Sesamol-deficiency in sesame was associated with the deletion of four C-terminal amino acids in CYP92B14. However, the thymine insertion generates a mutant that not only lacks the four C-terminal amino acids (Del4C) but also has a proline-serine replacement. One may speculate that the point mutation rather than the deletion is responsible for the inactivation of the CYP enzyme. Testing the activity of CYP92B14 containing serine as the fifth last amino acid may answer this question.

Yeast cells co-expressing CYP81Q1, CYP92B14 and CPR1 converted (+)-pinoresinol to (+)-sesaminol and (+)-sesamol via (+)-sesamin, demonstrating sequential lignan biosynthesis by the distinct P450s. This strain completely consumed (+)-sesamin, in contrast to cells expressing CYP92B14 and CPR1 only. Figure 4 illustrates the results obtained after 48 hours of incubation. To have an idea of the turnover rates and the efficiency of protein interaction, quantification of the substrate (+)-sesamin and the two products in the two yeast strains at various times between 0 h and 48 h may be helpful. Details: Line 49, C6-C3; line 113, what is the function of LOC105175177? Supplementary Tables, please place the caption at the top.

Reviewer #2 (Remarks to the Author):

The article by Murata et al. presents the discovery of the CYP450 responsible for the production of the lignans (+)-sesamol and (+)-sesaminol. It is an excellent piece of work which uses a broad range of experimental tools to unequivocally answer an important question. That said one part of the manuscript should be rewritten in my view. This is the part claiming a putative metabolon. I feel that this is not strong enough - the authors have suggestive evidence of protein co-operability here but not of metabolic channelling. I would suggest that they reword the corresponding sentences in the Results and Discussion since they are somewhat of a stretch and as such are not highly consistent with the rest of the manuscript which is meticulously carried out and soberly discussed. With this exception I found the manuscript highly convincing.

Responses to the comments of the reviewers:

Reviewer #1 (Remarks to the Author):

Sesame seeds contain (+)-sesamin, (+)-sesamol and (+)-sesaminol as the major lignans, which are interesting constituents due to their health-promoting activities. Here, the authors identify a new cytochrome P450 enzyme (CYP92B14), which converts (+)-sesamin to (+)-sesamol and (+)-sesaminol. CYP92B14 expression correlated with that of CYP81Q1 (sesamin synthase). A unique oxygenation scheme is proposed, based on the deuterium-labelling content of the products, as determined by NMR. CYP92B14 catalyses both direct oxygenation and oxidative rearrangement. The initial oxygenation step of α -oxy-substituted aryl groups may be widespread outside sesame lignans. The results reported are novel and interesting for a wider field and they will stimulate further research into this area.

Comment #1. *(+)-Sesamol-deficiency in sesame was associated with the deletion of four C-terminal amino acids in CYP92B14. However, the thymine insertion generates a mutant that not only lacks the four C-terminal amino acids (Del4C) but also has a proline-serine replacement. One may speculate that the point mutation rather than the deletion is responsible for the inactivation of the CYP enzyme. Testing the activity of CYP92B14 containing serine as the fifth last amino acid may answer this question.*

Response to the Comment #1: We thank the reviewer's comment and appreciate the idea of testing whether the substitution of C-terminal fifth last amino acid from proline to serine, rather than the deletion of C-terminal four amino acids, might be responsible for the loss of CYP92B14 activity. We generated an amino acid-substituted mutant of CYP92B14, P509S, and conducted enzyme activity assays. The result shows that P509S amino acid substitution of CYP92B14 does not result in the decrease of (+)-sesamol-producing activity. Therefore, we conclude that the lack of (+)-sesamol/(+)-sesaminol-producing activity in Del4C mutant is the result of the deletion of C-terminal 4 amino acids, rather than the amino acid substitution of P509. We added the data as **Supplementary Figure 23** and revised the manuscript as below.

Line 139: We added the following sentence in the manuscript; 'On the other hand, introduction of the amino acid substitution P509S to CYP92B14 did not result in the loss of the catalytic activity (**Supplementary Figure 23**).'.

Line 141: 'These data indicate that functional impairment, but not the lack of *CYP92B14* expression, is responsible for (+)-sesamol deficiency in #4294.' was revised to 'These data indicate that functional impairment originated from the lack of four amino acids in the C-terminus of CYP92B14, but not the lack of *CYP92B14* expression, is responsible for (+)-sesamol deficiency in #4294.'.

Line 245: Accordingly, the following sentences in Discussion ‘The C-terminal region of P450s is unlikely to be directly associated with the substrate-binding domain³⁵. However, the amino acid substitution P497R in *Homo sapiens* CYP27B1 leads to the functional impairment and is responsible for vitamin D-dependent rickets type I and cerebrotendinous xanthomatosis³⁶. The C-terminal region of CYP92B14 also contains a Pro residue (P509) that is substituted to Ser in Del4C due to the thymine insertion (**Fig. 2A and Supplementary Fig. 1**). Altogether, the requirement for a Pro and/or its adjacent residues in the C-terminal region of CYP92B14 for activity corroborates the implicit functional importance of the C-terminal region of P450³⁶’

was revised to

‘Although the C-terminal region of P450s is unlikely to be directly associated with the substrate-binding domain³⁵, the C-terminal region seems to be indispensable for the catalytic activity of P450. For example, an amino acid substitution P497R in *Homo sapiens* CYP27B1 leads to the functional impairment and is responsible for vitamin D-dependent rickets type I and cerebrotendinous xanthomatosis³⁶. On the other hand, in the case of CYP92B14, Del4C that lacks 4 amino acids in its C-terminal region and also carries the amino acid substitution P509S is unable to catalyze the reaction (**Fig. 2A and Supplementary Fig. 1**), whereas P509S mutant harboring full-length sequence remains active (**Supplementary Fig. 23**). Although the molecular mechanism behind the loss of oxygenation activity of these P450s is unclear, these data corroborate the implicit functional importance of the C-terminal region of P450³⁶’.

Comment #2. *Yeast cells co-expressing CYP81Q1, CYP92B14 and CPR1 converted (+)-pinoresinol to (+)-sesaminol and (+)-sesamol via (+)-sesamin, demonstrating sequential lignan biosynthesis by the distinct P450s. This strain completely consumed (+)-sesamin, in contrast to cells expressing CYP92B14 and CPR1 only. Figure 4 illustrates the results obtained after 48 hours of incubation. To have an idea of the turnover rates and the efficiency of protein interaction, quantification of the substrate (+)-sesamin and the two products in the two yeast strains at various times between 0 h and 48 h may be helpful.*

Response to the Comment #2: We thank the comment of the reviewer. According to the reviewer’s comment, we quantitated the production of (+)-sesamol and (+)-sesaminol versus the consumption of the substrate (+)-sesamin at various time points. (+)-Sesamol-producing activity of yeast cells expressing CYP92B14, CYP81Q1 and CPR1 were tested and the results were presented in **Supplementary Figure 23**.

Comment #3. *Details: Line 49, C6-C3; line 113, what is the function of LOC105175177? Supplementary Tables, please place the caption at the top.*

Response to the Comment #3: According to the reviewer's comment, we revised the manuscript as below;

Line 50: We thank the comment from the reviewer. We removed '(C6-C3)' from the sentence, since the information on carbon connectivity in the core structures of phenylpropanoids does not necessarily help the understanding of lignan biosynthesis.

Line 231: It is of particular interest whether the putative P450 enzyme encoded by LOC105175177 exhibits any enzyme activity towards sesame lignans. However, our attempts to identify the activity of the P450 enzyme have not been successful so far, since we failed to express LOC105175177 in yeast (**Supplementary Figure 22**). Future establishment of the expression system of LOC105175177 and its functional characterization in comparison to CYP92B14 may provide further insights into the molecular mechanism of ORA that is mediated by this type of P450 enzymes.

For Supplementary Tables, we placed the captions at the top, according to the reviewer's comment.

Line 250 in Supplementary Information: Supplementary Table 1: List of oligonucleotides used for RT-PCR and qPCR analyses.

Line 273 in Supplementary Information: Supplementary Table 2: Lignan content of seeds from sesame strains and their F1 plants.

Line 285 in Supplementary Information: Supplementary Table 3: Inheritance of the sesamol-in-deficient trait in F6 RILs resulting from the cross between #4294 and ITCFA2002.

Reviewer #2 (Remarks to the Author):

The article by Murata et al. presents the discovery of the CYP450 responsible for the production of the lignans (+)-sesamol and (+)-sesaminol. It is an excellent piece of work which uses a broad range of experimental tools to unequivocally answer an important question. That said one part of the manuscript should be rewritten in my view. This is the part claiming a putative metabolon. I feel that this is not strong enough - the authors have suggestive evidence of protein co-operability here but not of metabolic channelling. I would suggest that they reword the corresponding sentences in the Results and Discussion since they are somewhat of a stretch and as such are not highly consistent with the rest of the manuscript which is meticulously carried out and soberly discussed. With this exception I found the manuscript highly convincing.

Response to the Comment:

We deeply appreciate the valuable comments from the reviewer. Currently we do not have direct evidences to sufficiently show that CYP81Q1 and CYP92B14 are physically interacted and metabolically ‘channeled’. We therefore avoided the use of the word ‘channel’ in the submitted manuscript. Instead, we would like to focus on the result that the co-expression of CYP81Q1 tailors the activity of CYP92B14 in a cooperative manner. We believe this is quite valuable, since the data implies that the synergistic effect of multiple enzymes contribute, at least in part, to obtaining maximum efficiency in converting (+)-sesamin to (+)-sesamol (Fig. 3). To further support the notion, we performed time course experiments using yeast cells expressing CYP92B14, CPR1 and CYP81Q1, and added the data to the manuscript (**Supplementary Figure 23**). The data is noteworthy in that the co-expression of CYP81Q1 with CYP92B14 increases the production of (+)-sesamol throughout the experiments. From these results, while we agree with the reviewer to revise the manuscript by removing the term ‘metabolons’ in **Line 288**, we would like to keep the discussion on the positive effect of the co-expression of CYP81Q1 to the enzyme activity of CYP92B14. Please see the revised manuscript (**Line 286**) and data (**Supplementary Figure 23**).

Line 296: ‘Therefore, our data that the activity of CYP92B14 is functionally associated with CYP81Q1 provide unique opportunity to study structure-function relationship of enzymes that putatively form metabolons.’ was revised to ‘Therefore, our data that the activity of CYP92B14 is functionally associated with CYP81Q1 provide unique opportunity to study structure-function relationship of enzymes that function cooperatively.’.

Additional remarks:

In addition to the points listed above, we revised the submitted manuscript according to the guidelines in 'Nature Communications Manuscript Checklist', and also corrected minor errors. All these revisions are basically typographic, and do not affect the conclusions of the manuscript.

Revisions in the Main Manuscript

Line 33: We changed the tense of the results of the current study in the Abstract to present tense, according to the Manuscript Checklist. The points of revision are highlighted in yellow.

Submitted manuscript (150 words)

(+)-Sesamin, (+)-sesamolin and (+)-sesaminol glucosides are phenylpropanoid-derived specialized metabolites called lignans, and are rich in sesame (*Sesamum indicum*) seed. Despite their renowned anti-oxidative and health-promoting properties, the biosynthesis of (+)-sesamolin and (+)-sesaminol remained largely elusive. Here we showed that (+)-sesamolin-deficiency in sesame was genetically associated with the deletion of four C-terminal amino acids (Del4C) in a P450 enzyme CYP92B14 that constituted a novel clade separate from sesamin synthase CYP81Q1. Recombinant CYP92B14 converted (+)-sesamin to (+)-sesamolin and, unexpectedly, (+)-sesaminol through a novel oxygenation scheme designated as oxidative rearrangement of α -oxy-substituted aryl groups (ORA). Intriguingly, CYP92B14 also generated (+)-sesaminol through direct oxygenation of the aromatic ring. The activity of CYP92B14 was enhanced when co-expressed with CYP81Q1, implying functional coordination of CYP81Q1 with CYP92B14. The discovery of CYP92B14 not only uncovers the last steps in sesame lignan biosynthesis but highlights the remarkable catalytic plasticity of P450s that contributes to metabolic diversity in nature.

Revised Manuscript (150 words)

(+)-Sesamin, (+)-sesamolin and (+)-sesaminol glucosides are phenylpropanoid-derived specialized metabolites called lignans, and are rich in sesame (*Sesamum indicum*) seed. Despite their renowned anti-oxidative and health-promoting properties, the biosynthesis of (+)-sesamolin and (+)-sesaminol remained largely elusive. Here we **show** that (+)-sesamolin-deficiency in sesame **is** genetically associated with the deletion of four C-terminal amino acids (Del4C) in a P450 enzyme CYP92B14 that **constitutes** a novel clade separate from sesamin synthase CYP81Q1. Recombinant CYP92B14 **converts** (+)-sesamin to (+)-sesamolin and, unexpectedly, (+)-sesaminol through a novel oxygenation scheme designated as oxidative rearrangement of α -oxy-substituted aryl groups (ORA). Intriguingly, CYP92B14 also **generates** (+)-sesaminol through direct oxygenation of the aromatic ring. The activity of CYP92B14 **is** enhanced when co-expressed with CYP81Q1, implying functional coordination

of CYP81Q1 with CYP92B14. The discovery of CYP92B14 not only uncovers the last steps in sesame lignan biosynthesis but highlights the remarkable catalytic plasticity of P450s that contributes to metabolic diversity in nature.

Line 77: According to ‘*Nature Communications* Manuscript Checklist’, we added a paragraph briefly summarizing the results presented in this study.

Through genetic, genomic and biochemical approaches, here we identify a P450 monooxygenase CYP92B14 as the enzyme that is responsible for the oxygenation of (+)-sesamin to form (+)-sesamol and (+)-sesaminol. Furthermore, the labeling experiments using stable isotopes indicate that the oxygenation involves a novel reaction scheme designated as oxidative rearrangement of α -oxy-substituted aryl groups (ORA). The identification of CYP92B14 completes the biosynthesis of major lignans in sesame, and provides insights into yet to be characterized mechanisms of enzymatic oxygenation in specialized metabolites that possess heteroatoms at the α -position of alkyl-substituted aryl groups.

Line 89: A subheading ‘*Genome-based analysis of sesamol-deficient sesame accession #4294*’ was revised to ‘*Genetic analysis of sesamol-deficient accession #4294*’, in order to meet the requirement of ‘*Nature Communications* Manuscript Checklist’.

Line 101: We removed a word ‘from’ from the sentence, to revise the typographic error.

Line 108: A subheading ‘*Identification of CYP92B14 responsible for (+)-sesamol biosynthesis*’ was revised to ‘*CYP92B14 is responsible for (+)-sesamol biosynthesis*’, in order to meet the requirement of ‘*Nature Communications* Manuscript Checklist’.

Line 137: We added Supplementary Figure 22 according to the guideline of ‘*Nature Communications* Manuscript Checklist’.

Line 148: We revised the subtitle ‘*CO-expression of CYP81Q1 ameliorated catalytic activity of CYP92B14*’ to ‘*Effects of co-expression of CYP81Q1 on CYP92B14*’

Line 226: ‘Moreover’ was corrected to ‘Furthermore’.

Line 355: ‘(+)-Sesamin (ChromaDex) was purified by normal or reverse phase column chromatography.’ was revised to ‘(+)-Sesamin (ChromaDex) was sequentially purified by normal, followed by reverse phase column chromatography.’

Line 365: We added the following reference.

Kurita, T. *et al.* Efficient and convenient heterogeneous palladium-catalyzed regioselective deuteration at the benzylic position. *Chem. - A Eur. J.* **14**, 664–673 (2008).

Line 379: We believe 2,2'-³H₂-sesamin was mis-annotated as 7,7'-³H₂-sesamin in the reference 51 (Kato *et al.* (1998)). To avoid confusion, we removed part of the sentence as follows; 'Synthesis of (+)-2,2'-²H₂-sesamin was performed basically as reported previously⁵¹'.

Line 445: We revised the sentence in order to meet the requirement of '*Nature Communications* Manuscript Checklist' by adding a paragraph that describes the data availability in line 535.

Submitted manuscript

The remaining reads were assembled using Trinity (February 25, 2013 release) by normalization with maximum coverage to 30. The resulting fastq files are available through NCBI SRA (<http://www.ncbi.nlm.nih.gov/sra>; SRA Accession ID: PRJNA350858). For each sample, the Fragments per Kilobase Megareads (FPKM) values were calculated for contigs containing cDNA sequences responsible for CYP92B14, CYP81Q1 and CPR1 (DDBJ ID: LC209223) using Bowtie.

Revised manuscript

The remaining reads were assembled using Trinity (February 25, 2013 release) by normalization with maximum coverage to 30. For each sample, the Fragments per Kilobase Megareads (FPKM) values were calculated for contigs containing cDNA sequences responsible for CYP92B14, CYP81Q1 and CPR1 using Bowtie.

Line 545: We added a paragraph 'Data availability', according to '*Nature Communications* Manuscript Checklist'.

Data availability

The datasets generated and analyzed during the current study are available in the public repositories as follows; fastq files of RNASeq data at NCBI SRA (<http://www.ncbi.nlm.nih.gov/sra>; SRA Accession ID: PRJNA350858), gene identifiers LC199944 for CYP92B14 and LC209223 for CPR1.

Revisions in Supplementary Information

Line 34: '1,523 bp' was corrected to '1,525 bp'.

Line 35: 'S. indicum' was corrected to '*S. indicum*'.

Line 38: 'P508S' was corrected to 'P509S'.

Line 50: The number of replicates and the description of the error bars were added to the manuscript. 'The reaction was performed in triplicates. SD are shown by the error bars.'

Line 54: The colors of the graph in Supplementary Figure 3 were modified to avoid confusion for color-blinded readers.

Line 215: A blot of the whole gel of Figure 3C was added as Supplementary Figure 22 according to '*Nature Communications* Manuscript Checklist'.

Reviewers' comments:

Reviewer #1 (Remarks to the Author):

The authors have carried out additional experiments, as suggested by the reviewers. Furthermore, they have accurately revised their manuscript. All points raised in the reviews have been satisfactorily addressed.

Reviewer #2 (Remarks to the Author):

I am fully satisfied with the revisions that the authors made. I feel this will make a great contribution to plant natural product biology.

Reviewer #3 (Remarks to the Author):

This manuscript has been revised in response to a previous set of reviews, which were understandably quite complimentary. The authors report here a very thorough study leading to identification of a cytochrome P450 from sesame that produces both sesaminol and sesamol from sesamin, CYP92B14. The data for this is quite convincing, and the included labeling studies offer clear insight into the unusual production of sesamol. However, the suggested mechanism is arguably a bit too simplistic. It is unclear how C-C bond scission actually occurs in the mechanism presented in Figure 5, and the indicated stable intermediates seem a bit unlikely. In particular, CYPs utilize radical chemistry, and it would seem worthwhile considering how this might lead to the observed products. For example, it seems that these all could be the results of the initial generation of the radical at C1 rather than subsequent formation of a hydroxyl there (or at C6) – particularly given the potential for delocalization of the radical in the aromatic ring. This would not detract from the uniqueness of the catalyzed reaction, but would be much more mechanistically likely, and needs to be at least discussed if not also presented in Figure 5.

The studies also need a bit more clarification. Figure 3 is described as results from yeast microsomal assays. However, such assays are not described in the methods section. Preparation of yeast microsomes is not fully described. In particular, how much (extraction) buffer were the cells resuspended in? Also, microsomal preparations usually involve an initial low-speed spin to remove cell wall debris and other large/dense matter before the high-speed spin that pellets the microsomes, but this is not mentioned here. While not necessary for such preparation, the authors should confirm if that did not do this (and, hence, their preparations contain this 'other' material).

Minor corrections:

Line 161/2: please note here that both CYP81A1 and CYP92B14 were co-expressed with the requisite CPR1 redox partner.

Line 169: replace "required" with "incorporated oxygen from", as water is always present.

Line 244: replace "a Pro and/or its adjacent residues in the C-terminal region" with "the last four residues of the C-terminus", given the mutational (P508S) data now included.

Line 277: replace "structurally remote" with "phylogenetically distinct", as these CYPs will almost certainly share the same structural fold.

Line 286/7: replace "data that" with "data suggesting that" and "CYP81Q1 provide unique opportunity" with "CYP81Q1 may provide a novel system", given previous work on similar CYP-CYP interactions.

Figure 5: in 2,2' labeling scheme replace "3,6,2'-" with something that indicates the sesaminol has the second deuterium at either 3 or 6, perhaps "(3 or 6),2'-"

Responses to the comments of the reviewers:

Reviewer #1 (Remarks to the Author):

The authors have carried out additional experiments, as suggested by the reviewers. Furthermore, they have accurately revised their manuscript. All points raised in the reviews have been satisfactorily addressed.

Response to the Comment #1:

We would like to thank valuable comments from the reviewer #1 that help us improve the manuscript.

Reviewer #2 (Remarks to the Author):

I am fully satisfied with the revisions that the authors made. I feel this will make a great contribution to plant natural product biology.

Response to the Comment #2:

We appreciate the helpful advices to our manuscript.

Reviewer #3 (Remarks to the Author):

This manuscript has been revised in response to a previous set of reviews, which were understandably quite complimentary. The authors report here a very thorough study leading to identification of a cytochrome P450 from sesame that produces both sesaminol and sesamol from sesamin, CYP92B14. The data for this is quite convincing, and the included labeling studies offer clear insight into the unusual production of sesamol.

Comment #1. *However, the suggested mechanism is arguably a bit too simplistic. It is unclear how C-C bond scission actually occurs in the mechanism presented in Figure 5, and the indicated stable intermediates seem a bit unlikely. In particular, CYPs utilize radical chemistry, and it would seem worthwhile considering how this might lead to the observed products. For example, it seems that these all could be the results of the initial generation of the radical at C1 rather than subsequent formation of a hydroxyl there (or at C6) – particularly given the potential for delocalization of the radical in the aromatic ring. This would not detract from the uniqueness of the catalyzed reaction, but would be much more mechanistically likely, and needs to be at least discussed if not also presented in Figure 5.*

Response to the comment #1: We appreciate the thoughtful, inspiring comment of the reviewer. We revised the manuscript as follows;

Line 197: ‘CYP92B14 catalyzed the intricate reactions; a novel oxygenation designated as

oxidative rearrangement of α -oxy-substituted aryl groups (ORA) and a well-characterized direct oxygenation of aromatic ring. ORA involved i) the oxygenation of carbon atom on the aromatic ring system substituted with alkyl group, ii) the cleavage of the C-C bond between the aromatic and α -oxy-substituted alkyl groups, followed by iii) the addition of the generated phenol adduct and the oxonium intermediate (**Fig. 5B**).’ was revised to ‘CYP92B14 catalyzed the intricate reactions; a novel oxygenation designated as ORA and a well-characterized direct oxygenation of aromatic ring. ORA involved i) the oxygenation of carbon atom on the aromatic ring system substituted with alkyl group, possibly via radical intermediate, ii) the cleavage of the C-C bond between the aromatic and α -oxy-substituted alkyl groups, followed by iii) the addition of the generated phenol adduct and the oxonium intermediate (**Fig. 5B**).’

Line 202: The following sentences ‘The C-terminal region of CYP92B14 also contains a Pro residue (P509) that is substituted to Ser in Del4C due to the thymine insertion (**Fig. 2A and Supplementary Fig. 1**). However, the amino acid substitution of P509 to Ser (P509S) did not result in the loss of CYP92B14 activity (**Supplementary Fig. 23**). Therefore, the lack of 4 amino acids in the C-terminal region, rather than the substitution of P509S, is required for CYP92B14 to exhibit enzymatic activity. These data corroborate the implicit functional importance of the C-terminal region of P450.’ were removed and merged to the sentences ‘On the other hand, in the case of CYP92B14, Del4C that lacks the last four residues in its C-terminus and contains P509S substitution because of the thymine insertion, is unable to catalyze the reaction (**Fig. 2A and Supplementary Fig. 1**), whereas P509S mutant harboring full-length sequence remains active (**Supplementary Fig. 23**). Although the molecular mechanism behind the loss of oxygenation activity of these P450s is unclear, these data corroborate the implicit functional importance of the C-terminal region of P45036.’ in **Line 244** in the latest version of the manuscript.

Line 209: ‘Enzyme assays using deuterated (+)-sesamin clearly indicated that CYP92B14 catalyzed the oxidation of aromatic ring of (+)-sesamin, although it is unclear whether the initial oxidation of (+)-sesamin to generate intermediate I and III was achieved either by the concerted epoxidation or by the electrophilic attack of FeO complex²¹.’ was revised to ‘Enzyme assays using deuterated (+)-sesamin clearly indicated that CYP92B14 catalyzed the oxidation of aromatic ring of (+)-sesamin. P450 monooxygenases in general operate the initial oxidation of a phenolic substrate through electrophilic attack of FeO complex to the aromatic ring, which generates a cation rather than radical intermediate. Therefore, CYP92B14 likely generates cation intermediates I and III in the initial oxidation of (+)-sesamin.’

Accordingly, the reference #21

‘Guengerich, F. P. Common and Uncommon Cytochrome P450 Reactions Related to Metabolism and Chemical Toxicity. *Chem. Res. Toxicol.* 14, 611–650 (2001).’ was replaced to ‘Tomberg, A., Pottel, J., Liu, Z., Labute, P. & Moitessier, N. Understanding P450-mediated Bio-transformations into Epoxide and Phenolic Metabolites. *Angew. Chemie - Int. Ed.* 54,

13743–13747 (2015).’

Line 216: ‘ORA is a quite unique oxygenation scheme, since plant specialized metabolites rarely harbor an insertion of an oxygen atom between aromatic and α -oxy-substituted alkyl group as in (+)-sesamolin¹¹. However, the initial oxygenation step of ORA might be widespread outside sesame lignans.’ was revised to ‘Although plant specialized metabolites rarely harbor an insertion of an oxygen atom between aromatic and α -oxy-substituted alkyl group as in (+)-sesamolin¹¹, the initial oxygenation step of ORA might be widespread outside sesame lignans.’.

We also revised **Fig. 5B** in order to better explain the revised manuscript by describing the reaction intermediates as heme-iron complexes.

Comment #2. *The studies also need a bit more clarification. Figure 3 is described as results from yeast microsomal assays. However, such assays are not described in the methods section. Preparation of yeast microsomes is not fully described. In particular, how much (extraction) buffer were the cells resuspended in? Also, microsomal preparations usually involve an initial low-speed spin to remove cell wall debris and other large/dense matter before the high-speed spin that pellets the microsomes, but this is not mentioned here. While not necessary for such preparation, the authors should confirm if that did not do this (and, hence, their preparations contain this ‘other’ material).*

Response to the comment#2: We deeply appreciate the comment and would like to revise the respective sections of the manuscript as follows;

Line 454: We moved a paragraph ‘*Enzyme assays using sesame microsome fractions*’ from Line 466 to Line 454.

Line 483: A paragraph ‘*Enzyme assays using yeast microsome fractions*’ was added according to the reviewer’s comment. ‘Yeast cell lines expressing CYP92B14 and CPR1, Del4C and CPR1, CYP92B14 alone or CPR1 alone were precultured in 50 ml SD medium lacking respective amino acids overnight at 120 rpm 30°C. The saturated culture was transferred to 950 ml SD liquid medium without respective amino acids and cultured at 120 rpm 30°C until the OD₆₀₀ reached 2.0. The cells were harvested by centrifugation at 10,000 x g for 15 min at 4°C. The pellet was rinsed once and resuspended in 10 ml of Extraction buffer I, followed by sonication at 4°C. The homogenate was centrifuged at 21,000 x g for 10 min at 4°C, then the supernatant was ultracentrifuged at 70,000 x g for 1 hr at 4°C. The pellet was resuspended in 600 μ l of Extraction Buffer I and subjected to enzyme assays. The microsome fraction (50 μ l) was mixed with 100 μ M sesamin and 1 mM NADPH in a final volume of 100 μ l and incubated for 60 min at 37°C. The reaction was terminated by the addition of an equal volume of acetonitrile, and the mixture was filtered through a Millex-LH syringe filter (Merck Millipore),

then subjected to HPLC analysis.’

Line 536: We revised the sentence ‘The cells were resuspended in Extraction buffer I containing 100 mM Tris-Cl (pH 8.0), 0.1% β -mercaptoethanol, 20% glycerol and 1x Complete EDTA-free protease inhibitor (Roche), disrupted by sonication on ice, then the homogenate was ultracentrifuged at 75,000 x g for 1 hr at 4°C.’ to ‘The cells were resuspended in Extraction buffer I, disrupted by sonication on ice, then the homogenate was centrifuged at 10,000 x g for 15 min at 4°C, followed by ultracentrifugation at 75,000 x g for 1 hr at 4°C.’

Minor corrections:

Line 161/2: *please note here that both CYP81A1 and CYP92B14 were co-expressed with the requisite CPR1 redox partner.*

Response to the comment: The corresponding part of the sentence was accordingly revised to ‘it was completely consumed in yeast cells expressing CYP92B14 and CYP81Q1 together with CPR1.’

Line 169: *replace “required” with “incorporated oxygen from”, as water is always present.*

Response to the comment: According to the reviewer’s comment, we revised the part of the sentence from ‘CYP92B14 required molecular oxygen,’ to ‘CYP92B14 incorporated oxygen from molecular oxygen,’.

Line 244: *replace “a Pro and/or its adjacent residues in the C-terminal region” with “the last four residues of the C-terminus”, given the mutational (P508S) data now included.*

Response to the comment: We thank the comment from the reviewer. We revised the sentence, which is in **Line 244** in the latest version of the revised manuscript, as follows; ‘On the other hand, in the case of CYP92B14, Del4C that lacks the last four residues in its C-terminus is unable to catalyze the reaction (**Fig. 2A and Supplementary Fig. 1**), whereas P509S mutant harboring full-length sequence remains active (**Supplementary Fig. 23**).’.

Line 277: *replace “structurally remote” with “phylogenetically distinct”, as these CYPs will almost certainly share the same structural fold.*

Response to the comment: We appreciate the comment and revised the corresponding sentence which is in **Line 282** in the latest version of the manuscript.

Line 286/7: *replace “data that” with “data suggesting that” and “CYP81Q1 provide unique opportunity” with “CYP81Q1 may provide a novel system”, given previous work on similar*

CYP-CYP interactions.

Response to the comment: We agree with the comment from the reviewer, and revised the manuscript accordingly. The sentence is in **Line 291** in the latest version of the manuscript.

Figure 5: *in 2,2' labeling scheme replace "3,6,2'-" with something that indicates the sesaminol has the second deuterium at either 3 or 6, perhaps "(3 or 6),2'-"*

Response to the comment:

We appreciate the comment and revised the sentence accordingly.

Additional Remarks:

Line 197: 'CYP92B14 catalyzed the intricate reactions; a novel oxygenation designated as oxidative rearrangement of α -oxy-substituted aryl groups (ORA) and a well-characterized direct oxygenation of aromatic ring.' was corrected to 'CYP92B14 catalyzed the intricate reactions; a novel oxygenation designated as ORA and a well-characterized direct oxygenation of aromatic ring.'

Line 457: '100 μ M 2 H-labeled sesamin' was corrected to '100 μ M sesamin'. We apologize for the typo error.

Legend for Fig. 5: 'White letters indicate the molar ratio of respective enzyme assay products deduced from NMR analysis.' was revised to 'White letters indicate the molar ratio of respective enzyme assay products deduced from LC and NMR analysis.'

REVIEWERS' COMMENTS:

Reviewer #3 (Remarks to the Author):

The authors have satisfactorily addressed my concerns.

Responses to the comments of the reviewers:

Reviewer #3 (Remarks to the Author):

The authors have satisfactorily addressed my concerns.

Response to the Comment from Reviewer #3:

We would like to appreciate valuable comments from the reviewer #3.

Additional remarks:

We changed the order of Supplementary Figures and revised the manuscript accordingly.

In order to obtain high-resolution figures, we re-analyzed the LC-MS data in Supplementary Figure 8. The revised data consistently support the view that CYP92B14 oxygenates (+)-sesamin using molecular oxygen as an oxygen donor.